# Recognition of Water Colour Anomaly by Using Hue Angle and Sentinel 2 Image

**Yelong Zhao** [1,2]**, Qian Shen** [2,]*****, Qian Wang** [3]**, Fan Yang** [1]**, Shenglei Wang** [2,4]**, Junsheng Li** [2] **, Fangfang Zhang** [2] **and Yue Yao** [2]

1. Liaoning Technical University, Fuxin 123000, China; 471710035@lntu.edu.cn (Y.Z.); yangfan_cehui@lntu.edu.cn (F.Y.)
2. Key Laboratory of Digital Earth Science, Aerospace Information Research Institute, Chinese Academy of Sciences, Beijing 100049, China; wangsl@radi.ac.cn (S.W.); lijs@radi.ac.cn (J.L.); zhangff07@radi.ac.cn (F.Z.); yaoyue@aircas.ac.cn (Y.Y.)
3. Ministry of Ecology and Environment of the People's Republic of China, Beijing 100006, China; wfsyysc@mee.gov.cn
4. PKU Institute of Remote Sensing and Geographical Information System, Peking University, Beijing 100871, China
* Correspondence: shenqian@radi.ac.cn; Tel.: +86-010-8217-8181

**Abstract:** As polluted water bodies are often small in area and widely distributed, performing artificial field screening is difficult; however, remote-sensing-based screening has the advantages of being rapid, large-scale, and dynamic. Polluted water bodies often show anomalous water colours, such as black, grey, and red. Therefore, the large-scale recognition of suspected polluted water bodies through high-resolution remote-sensing images and water colour can improve the screening efficiency and narrow the screening scope. However, few studies have been conducted on such kinds of water bodies. The hue angle of a water body is a parameter used to describe colour in the International Commission on Illumination (CIE) colour space. Based on the measured data, the water body with a hue angle greater than 230.958° is defined as a water colour anomaly, which is recognised based on the Sentinel-2 image through the threshold set in this study. The results showed that the hue angle of the water body was extracted from the Sentinel-2 image, and the accuracy of the hue angle calculated by the in situ remote-sensing reflectance $R_{rs}(\lambda)$ was evaluated, where the root mean square error (RMSE) and mean relative error (MRE) were 4.397° and 1.744%, respectively, proving that this method is feasible. The hue angle was calculated for a water colour anomaly and a general water body in Qiqihar. The water body was regarded as a water colour anomaly when the hue angle was >230.958° and as a general water body when the hue angle was ≤230.958°. High-quality Sentinel-2 images of Qiqihar taken from May 2016 to August 2019 were chosen, and the position of the water body remained unchanged; there was no error or omission, and the hue angle of the water colour anomaly changed obviously, indicating that this method had good stability. Additionally, the method proposed is only suitable for optical deep water, not for optical shallow water. When this method was applied to Xiong'an New Area, the results showed good recognition accuracy, demonstrating good universality of this method. In this study, taking Qiqihar as an example, a surface survey experiment was conducted from October 14 to 15, 2018, and the measured data of six general and four anomalous water sample points were obtained, including water quality terms such as $R_{rs}(\lambda)$, transparency, water colour, water temperature, and turbidity.

**Keywords:** Sentinel-2; remote sensing recognition; water colour anomaly; the hue angle of water body; remote sensing reflectance

## 1. Introduction

Water colour anomaly usually indicates that, when the water body is polluted, the water colour can indirectly reflect the quality of the water body. The anomalous colours of a water body usually include red, black, and grey, but not green, which is the colour of a eutrophic water body. Several uncertain factors, such as the difference in colour sensitivity between individuals and the experience of water-colour recognition, are considered to recognise water colour anomaly through artificial visual interpretation. To avoid the impacts of these uncertain factors, this study adopted the hue angle of a water body to recognise water colour anomaly, and we confirmed that the water colour was anomalous when the hue angle of the water body was greater than 230.958°. Water colour anomaly usually results from illegal discharge and dumping of pollutants, such as industrial wastewater, industrial solid waste, and domestic waste, leading to a significantly higher concentration of pollutants in the surface water than that in an ordinary natural water body; this imposes serious harm on aquatic organisms and the surrounding ecology and environment. The factors that lead to a water colour anomaly are generally divided into three categories: (1) the industrial category, which comprises waste acids, waste oils, heavy metals, refractory organic matter, and other industrial solid wastes; (2) the living category, which comprises domestic sewage, domestic waste, and other wastes; and (3) the agriculture category, comprising livestock and poultry wastewater and manure accumulation. Figure 1 presents examples of water colour anomalies [1–5].

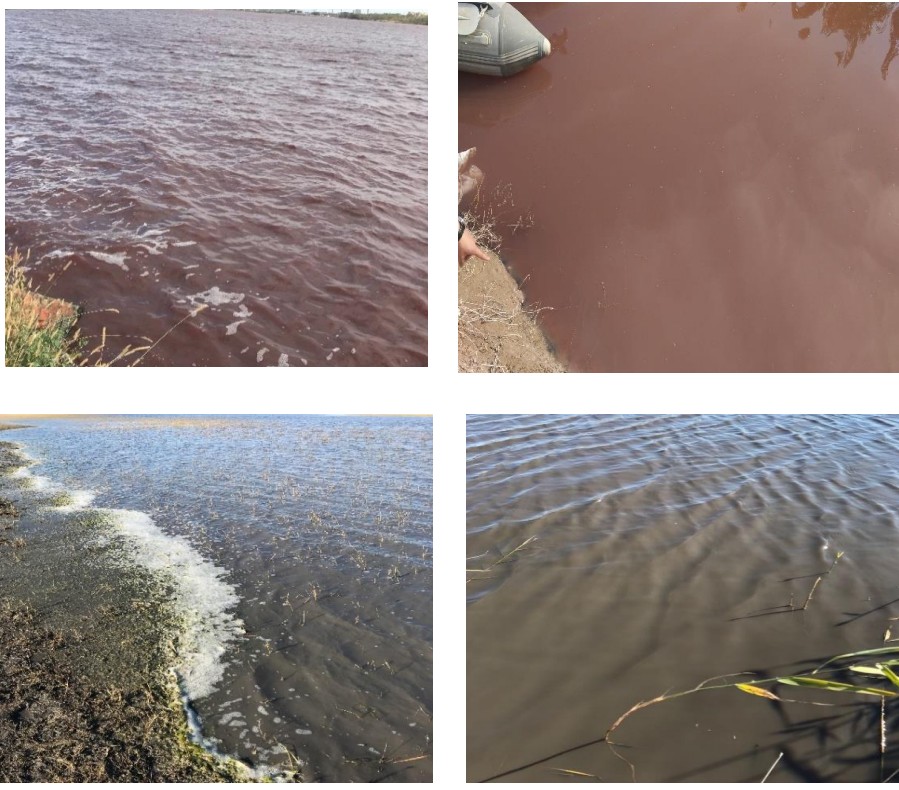

**Figure 1.** Illustrations of water colour anomalies.

The water colour anomaly is not distributed in a concentrated manner but over a wide range. The conventional manual field-survey method cannot perform fast and large-scale real-time dynamic monitoring, has the disadvantages of high labour and economic costs, low efficiency, and is prone to cause errors in reports. Conversely, remote-sensing technology has the advantages of a wide-monitoring range, fast information acquisition, and short cycle; it can therefore reflect the dynamics of the water body in real-time, as well as employ historical satellite data to explore the change of law of water colour over time.

In 1890, Francois Alphonse Forel [6], for the first time, put forward the standard for a colour classification of water bodies and divided blue and green water bodies into 11 colour grades. Initially, the 11 colours in the test tube of a water colour meter were created by adding different proportions of yellow agent (potassium chromate) to a blue-agent (copper sulphate) solution. In 1892, Willi-Ule added ten colours, from turquoise to reddish-brown, to the water colour meter. These 11 and ten colour grades by Forel and Ule are today known as the 21-colour Forel–Ule colourimeter (or Forel–Ule colour index; FUI). Novoa et al. [7] conducted spectral analysis on the FUI scale by preparing protocols, transmission assays, and through chromatography. In recent years, some progress has been made in monitoring water quality through remote-sensing technology and water colour. In 1978, Alfold and Munday [8] assessed the correlation between hue and colour saturation in an HIS colour space based on a Landsat MSS false-colour synthetic image and water quality. Wernand and van der Woerd [9] found that there is good consistency between the dominant wavelength in a International Commission on Illumination (CIE) colour space and FUI index. According to the International Commission on Illumination (CIE) colour space calculated through Landsat TM red, green, and blue bands, Jaquet and Zand [10] qualitatively evaluated the relationship between the trophic status of a water body and the coordinates of the CIE colour space. Bukata, Bruton, and Jerome [11] and Bukata et al. [12] determined the relationship between water colour and water quality. Dörnhöfer [13] used several years of medium-resolution-imaging spectrometer (MERIS) data and a neural network algorithm to retrieve the absorption of chlorophyll, a total suspended matter, and coloured dissolved organic matter (440 nm) in Lake Kummerow and examined the colour change in the lake. Based on moderate-resolution-imaging spectroradiometry (MODIS) data, Wang [14] classified water colours by retrieving the main wavelength of water colour and analysing the relationship between water colour and optically active components in a water body. Wernand [15] proposed the FUI-level algorithm of water body extraction based on MERIS-based remote-sensing images. Garaba et al. [16] applied the FUI system to classify natural waters. By using MODIS data, Li [17] explored the changes in water quality and the trophic status of China's top ten lakes from 2000 to 2012 by observing the changes in water colour. Shen [18] adopted the purity method to recognise black and odorous water bodies of rivers in cities on the strength of high-resolution multispectral images. Wernand et al. [9] performed spectral analysis using the Forel-Ule ocean-colour comparator scale. Rudorff et al. [19] applied a semi-analytical method to retrieve turbidity based on Landsat and MODIS-Aqua satellite data to measure the impact of mining wastewater disaster on the turbidity of the Doce River plume on the east coast of Brazil. Oron and Gitelson [20] employed remote sensing to monitor the water quality in a wastewater pond in real-time, determined the reasons for the change in remote-sensing reflectance $R_{rs}$ ($\lambda$) of the wastewater, and proposed to monitor the severely polluted water in real-time by observing the change in $R_{rs}$ ($\lambda$).

In the past, most studies on water-quality monitoring based on remote sensing employed satellite data of MERIS, MODIS, and Landsat to monitor the general water bodies of large lakes and reservoirs with normal water colours; however, few studies focused on the recognition and monitoring of water colour anomalies [13–16,19–23]. Owing to a long observation time, high spatial resolution, and the same spatial resolution of all bands (except for two thermal infrared bands), Landsat data are conducive to the long-term monitoring and analysis of water quality [21]. In contrast, MERIS and MODIS data are mostly applied to water-quality monitoring of large lakes and reservoirs owing to low spatial resolutions [13,19]. As water colour anomalies are mostly observed in pit ponds in urban rivers and industrial areas and cover a small area, the spatial resolution of the above-mentioned satellite data is not enough for recognising a water colour anomaly. In this paper, the hue angle of a water body was extracted from Sentinel-2 images to recognise water colour anomaly, which is applicable for water bodies with red, brown, and grey colours but not the eutrophic water bodies with green colours. The Sentinel-2 satellite has the advantages of high time resolution (Sentinel 2A and Sentinel 2B are complementary, and the revisit cycle is five days), high spatial resolution (10–60 m), a large width of 290 km × 290 km, and the satellite image does not require geometric correction owing to a stable orbit. To recognise water colour anomalies by extracting the hue angle of a water body from a Sentinel-2

image, it is necessary to conduct pre-processing, such as atmospheric correction, band resampling, and water body extraction.

Sen2Cor and SNAP software developed by the European Space Agency (ESA) were used for the atmospheric correction and band resampling of the Sentinel-2 image data to gain the surface reflectance data with a spatial resolution of 10 m. There are two main methods for extracting water body based on remote-sensing data: (1) through water body indices, such as modified normalised difference water index (MNDWI), normalised difference water index (NDWI), automated water extraction index (AWEI), multiband water index (MBWI), and multispectral water index (MuWI) [24–30], and (2) through classification methods, such as support vector machine (SVM), maximum likelihood method, decision tree, and random forest [31,32]. The advantages of using water body indices to extract water bodies include easy application and rapid results; however, they have limited band information, show large errors of water body extraction results, and have complex calculations of optimal thresholds, in which each scene image must match an optimal threshold. By comparison, the classification methods boast high accuracy; however, the collection of samples is complex, the processing is time-consuming, and the accuracy of sample selection can affect the final results of the water extractions.

## 2. Study Area and Data

### 2.1. Overview of the Study Area

Qiqihar City is located in the Heilongjiang Province, Songnen Plain in Northeast China, at longitude 122°–126° East and latitude 45°–48° North. In July 2018, a black and odorous water body was discovered in the sewage pit pond in the Ang'angxi District, Qiqihar City. The Heilongjiang Provincial Government issued a supervision letter to the Qiqihar Municipal Government, urging Qiqihar to address this problem of environmental pollution [33]. Before July 2016, the discharge of domestic sewage from the Ang'angxi District and the wastewater produced by industrial and mining companies in the city, such as by the Heilongjiang Fengyuan Industrial Group Co. Ltd., Ankang Bioengineering Co. Ltd., and the Huoliyuan Yeast Co. Ltd., formed several huge black and odorous water bodies [33]. The Xiong'an New Area is located in Baoding City, Hebei Province, China, the planning scope of which includes Xiong County, Rongcheng County, and Anxin County. In 2017, 387 pit ponds, resulting from years of accumulation of garbage and sewage, were screened in Anxin County, and a treatment plan of "one pit pond, one policy" [34] was formulated. The study area is shown in Figure 2.

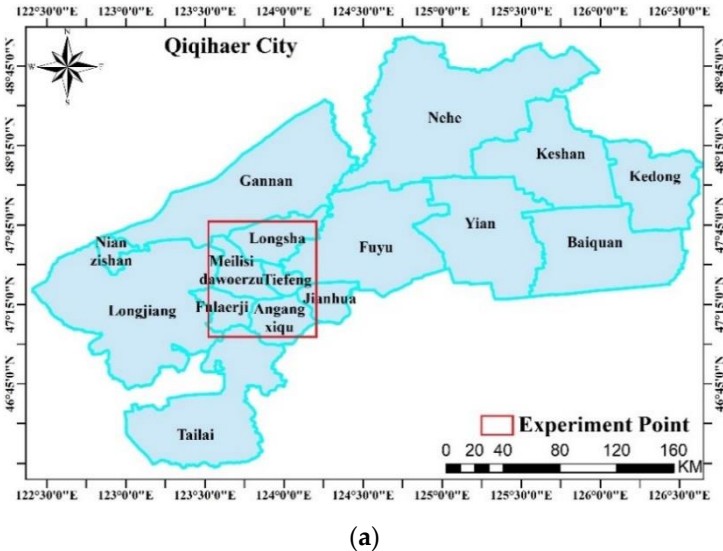

(a)

**Figure 2.** *Cont.*

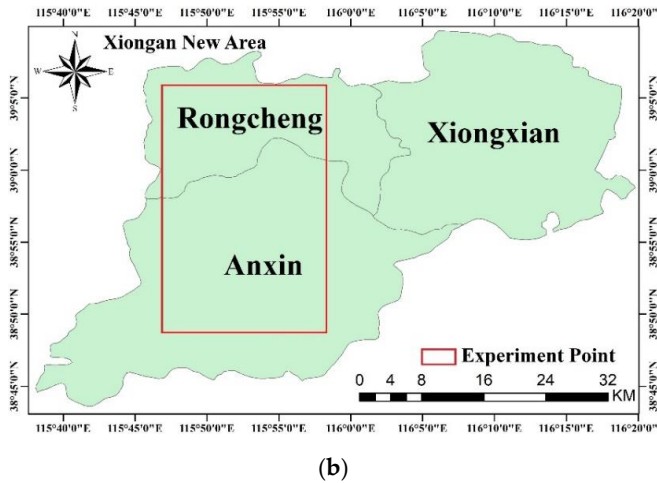

(**b**)

**Figure 2.** Study areas at (**a**) Qiqihaer City and (**b**) the Xiong'an New Area.

## 2.2. Data Acquisition

On October 14 and 15, 2018, the data of ten water samples were collected in the Fularji District, Ang'angxi District, Durbert County, and Fuyu County of Qiqihar City from six general water bodies and four water colour anomalies. On April 28, 2017, a total of six water samples were collected in Anxin County, Xiong'an New Area, from four general water bodies and two water colour anomalies. On October 5, 2018, the data of 13 water samples, including 12 general water bodies and one water colour anomaly, in Rongcheng County, Xiong'an New Area, were collected. At each test site, the remote-sensing reflectance, $R_{rs}$ (λ), of the water body was measured, photos were taken, and the in situ situation of the water body was recorded. In Qiqihar City and the Xiong'an New Area in 2018 and 2017, respectively, quasi-synchronous satellite-to-ground tests were conducted with a time difference of 10 days. In 2018, a satellite-ground synchronous test was performed in the Xiong'an New Area, as shown in Figure 3.

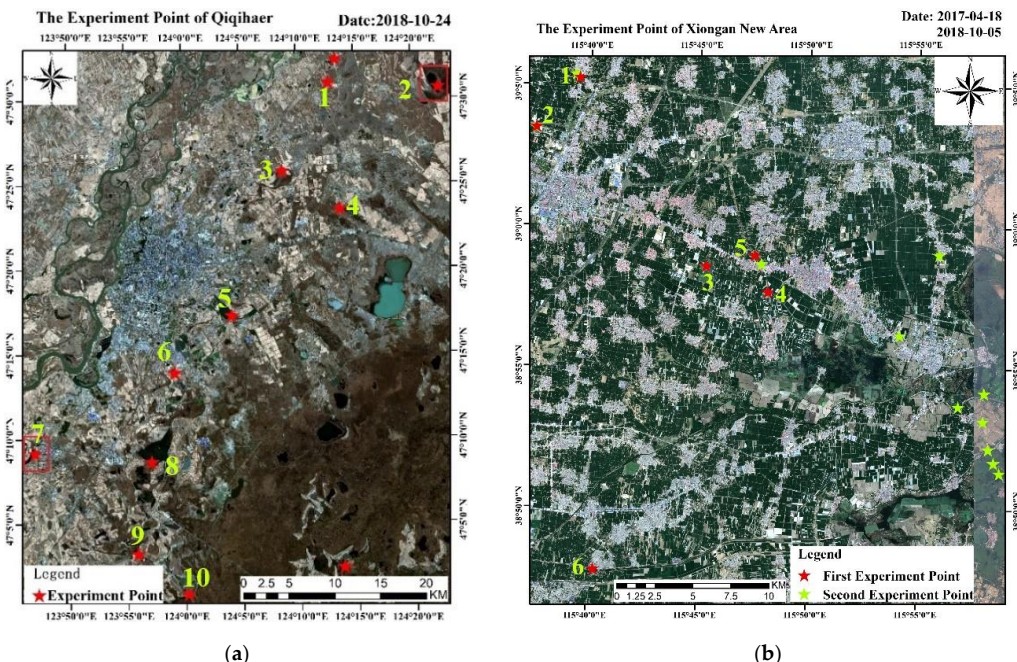

(**a**)　　　　　　　　　　　　　　　　　　　　　　　　　　　(**b**)

**Figure 3.** Test Sites at (**a**) Qiqihaer City and (**b**) the Xiong'an New Area. Note: In Qiqihar, samples 1-6 are general water bodies, and samples 7-10 are anomalous waters. In the Xiong'an New Area, samples 1-4 are general water bodies, and samples 5-6 are anomalous waters.

### 2.2.1. In Situ Remote-Sensing Reflectance

At each test site shown in Figure 3, the ASD Field-SpecR3 portable spectroradiometer was used to collect the water-surface spectra. In addition, spectral measurement was performed by adopting NASA Ocean Optics Protocols [35] and the water spectral measurement method proposed by Tang et al. [36]. To avoid the influences of solar flares and hull shadows, the observation azimuth and zenith angles were set at 135° and 40°, respectively [35,36]. The spectrum collection at each sampling point followed the order of reference board, water body, sky light, and reference board, in which ten spectra were collected from the reference board and sky light, and 15 spectra were collected from the water body. When calculating the remote-sensing reflectance, the outliers in the acquisition process were eliminated, and then, the mean value of the remaining spectra was calculated. The remote-sensing reflectance is calculated as:

$$R_{rs}(\lambda) = \frac{L_t(\lambda) \cdot r \times L_{sky}(\lambda)}{L_p(\lambda)/\rho_p \times \pi} \tag{1}$$

where $L_t(\lambda)$ is the upward radiance of the water body and $r$ is the reflectance of sky light at the air–water interface, the calculation results of which are determined by factors such as the sun position, observation geometry, wind speed, and wind direction. When the observation zenith angle is 40°, $r = 0.0245$, according to the Fresenel formula. $L_{sky}$ is the downward radiance of the sky light, $L_p(\lambda)$ is the radiance of the reference board, and $\rho_p$ is the standard reflectance of the reference board obtained through a laboratory calibration method.

### 2.2.2. Sentinel-2 Image Data and Pre-Processing

The Sentinel-2 satellite was launched by ESA in 2015 to provide global coverage and open optical-remote-sensing data. Compared with the images obtained by the Landsat series satellites similar to Sentinel 2, Sentinel-2 images have the advantages of higher spatial resolution, richer spectral bands, shorter revisit cycles, and larger width, thereby showing great application potential in water science research and other fields [37–43].

In this research, the Level 1C Top-Of-Atmosphere (L1C TOA) reflectance products were downloaded from the ESA website. The atmospheric correction and band resampling of L1C TOA reflectance products were performed using the Sen2Cor and SNAP software developed by ESA, and surface reflectance data with a spatial resolution of 10 m in each band were acquired. In addition, when screening the data, the cloud coverage was set as less than 20%, and no snow was recorded in the area. Under these conditions, 41 images of Qiqihar captured from May 2016 to August 2019 and 68 images of the Xiong'an New Area captured from April 2016 to August 2019 were selected. These periods were considered as Qiqihar and the Xiong'an New Area are located in Northeast China and North China, respectively, and the frost-free period in Northeast China is shorter than that in North China.

## 3. Study Methods

First, the water body was extracted from a Sentinel-2 image with a spatial resolution of 10 m based on the MuWI, after which the hue angle of the water body pixel was calculated, and then the suspected water colour anomaly was recognised in line with the threshold.

### 3.1. Accuracy Evaluation of Indices

Generally, the mean relative error (MRE) and root mean square error (RMSE) defined below are adopted when evaluating the accuracy of water body extractions:

$$MRE = \frac{1}{n} \frac{\sum_{1}^{n} |X - X'|}{X} \tag{2}$$

$$RMSE = \sqrt{\frac{\sum_1^n (X - X')^2}{n}} \tag{3}$$

when evaluating the accuracy of water body extractions, where X represents the true value, X' is the measured value, and n is the number of test areas. When evaluating the accuracy of the hue angle of the water body, however, then X represents the result of the hue angle calculated by the in situ remote-sensing reflectance $R_{rs}(\lambda)$, X' represents the result of the hue angle calculated by the remote-sensing reflectance $R_{rs}(\lambda)$ of the Sentinel-2 image, and n denotes the number of water sample points.

Regarding the recognition accuracy of the water colour anomaly, the recognition accuracy rate was evaluated as:

$$\text{Accuracy} = M/N \times 100\% \tag{4}$$

where N represents the total number of samples, and M represents the accurate number of models recognised by the proposed model.

### 3.2. Water Body Extraction

Water body indices, such as NDWI [24] and MNDWI [25], have good recognition effects on general and eutrophic water bodies; they are defined as follows:

$$NDWI = \frac{Green - NIR}{Green + NIR} \tag{5}$$

$$MNDWI = \frac{Green - SWIR}{Green + SWIR} \tag{6}$$

However, they cannot recognise and extract water bodies with high turbidity and anomalous colours (such as red, grey, and black water bodies). The final water extraction results show only clean and general water bodies but not highly turbid water bodies and water colour anomalies. Therefore, water colour anomalies cannot be recognised among these water bodies. To better recognise the general water body, highly turbid water body, and water colour anomaly, a water body in the study area, MuWI, which is especially designed for Sentinel-2 remote-sensing images, was selected to extract water bodies in the study area [28]. The MuWI was established through an SVM model, and the complete version of this index, namely MuWI-C, was obtained through SVM training, as follows:

$$\begin{aligned} \text{MuWI-C} = &-16.4ND(2,3) - 6.9ND(2,4) - 8.2ND(2,8) - 8.8ND(2,11) + 9.6ND(2,12) + \\ &10.8ND(3,8) + 6.1ND(3,11) + 13.6ND(3,12) - 0.28ND(4,8) - 3.9ND(4,11) - 2.1ND(4,12) - \\ &5.3ND(8,11) - 5.3ND(11,12) - 0.33 \end{aligned} \tag{7}$$

where ND $(i,j)$ represents the normalised difference between bands $i$ and $j$ in the Sentinel-2 image, expressed as

$$ND(i, j) = \frac{i - j}{i + j} \tag{8}$$

However, MuWI-C contains several terms, and the terms with high integer coefficient weights in MuWI-C were selected, and therefore, the MuWI-R expression was obtained:

$$\text{MUWI-R} = -4ND(2,3) + 2ND(3,8) + 2ND(3,12) - ND(3,11) \tag{9}$$

where $ND(2,3)$, $ND(3,8)$, and $ND(3,12)$ are three terms with the highest integer coefficient weights in MuWI-C. $ND(3,11)$ was retained to preserve the applicability of the two shortwave infrared bands for water pixels with high albedo. As the constant term of MUWI-R could not be obtained through the SVM model, it was omitted. By taking the surface reflectance data after atmospheric correction as input, MuWI-C and MuWI-R images were obtained. In general, the extraction of water bodies by

using the threshold method based on remote-sensing images is the most common extraction method. The optimal threshold is calculated using autonomous iteration or statistical methods; however, the calculation process is complex, and each scene image must match an optimal threshold, resulting in an overwhelming workload of obtaining the optimal threshold. To reduce the amount of calculation of the optimal threshold and more efficiently and automatically distinguish between water and non-water areas in the image, the K-means clustering analysis method was used in this study in the remote-sensing image, thus lessening the amount of calculations to obtain the optimal threshold or the errors caused by the artificial selection of empirical values [44]. To reduce the misextraction of the two types of water body indices and the impact of the water-land boundary, after obtaining the intersection of the result of MuWI-C and MuWI-R, a pixel was corroded inward based on the mathematical morphology, and the final water extraction result was obtained [45].

Figure 4 shows the extraction results of these three water body indices on the same water body in the same Sentinel-2 scene image, where A, B, and C are highly turbid water colour anomalies with black, red, and grey colours, respectively. These three methods can be employed to recognise the general water body with normal colour and the green water body with severe eutrophication; however, NDWI [24] and MNDWI [25] cannot recognise the highly turbid water body with anomalous colour. In contrast, the intersection of MuWI-R and MuWI-C can well recognise the highly turbid water body with anomalous colour. The two accuracy evaluation indices were only applied to evaluate the extraction results of MuWI; however, the accuracy evaluation of NDWI and MNDWI was not performed, because these indices cannot completely recognise the outline of turbid water bodies with water colour anomalies. By using a visual interpretation method, the water body was artificially outlined as the true value of the water body, and the MRE and RMSE of the intersection result of MuWI-C and MuWI-R were obtained as 0.04% and 7.48, respectively.

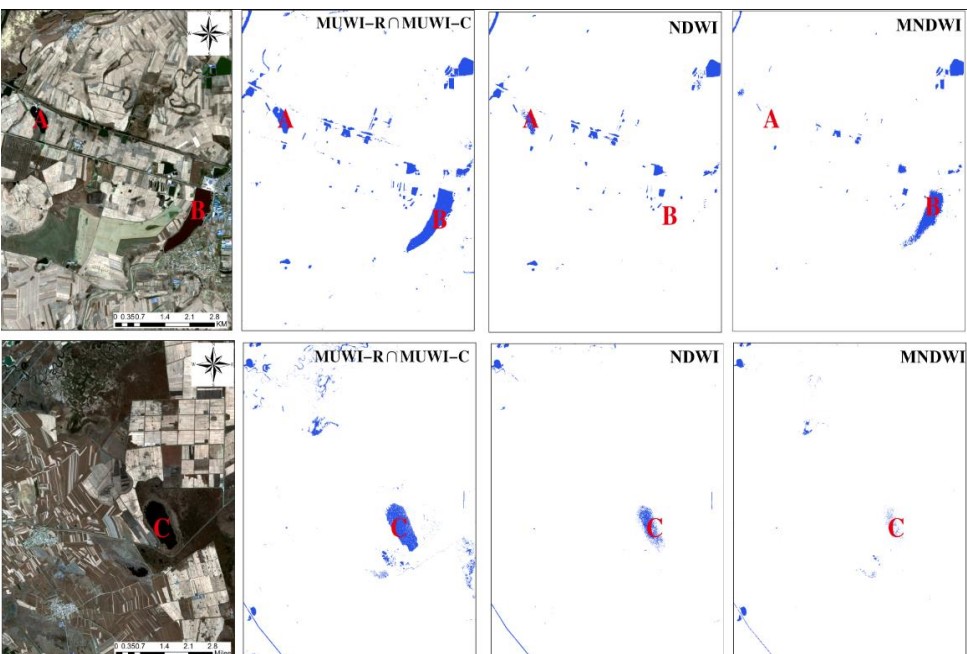

**Figure 4.** Sentinel-2 RGB image and corresponding water maps classified by spectral water indices, multispectral water index (MuWI)-R∩MuWI-C, modified normalised difference water index (MNDWI), and normalised difference water index (NDWI). Note: A, B, and C are the red box areas in Figure 3. (Location: Angxi District and Fuyu County of Qiqihar City).

## 3.3. Calculation of the Hue Angle of the Water Body

The CIE has stipulated a set of standard chromaticity systems to gain consistent measurement results [46]. The hue angle of the water body is a parameter used to describe colour in the CIE colour

space. The tristimulus values of X, Y, and Z of the CIE in the three bands of R, G, and B of the Sentinel 2 were calculated using the RGB conversion method [13,16], as follows:

$$X = 2.7689R + 1.7517G + 1.1302B$$
$$Y = 1.0000R + 4.5907G + 0.0601B \quad (10)$$
$$Z = 0.0000R + 0.0565G + 5.5934B$$

The chromaticity coordinates of the CIE were obtained by X, Y, and Z, and the calculated hue coordinates were normalised to 0–l. Thus, a new coordinate system was obtained based on the chromaticity coordinates:

$$x = X/(X+Y+Z)$$
$$y = Y/(X+Y+Z) \quad (11)$$
$$z = Z/(X+Y+Z)$$

As shown in Figure 5, the hue angle $\alpha$ was calculated by $(x', y')$ in the CIE chromaticity diagram as

$$\alpha = \text{ARCTAN2}(y',x') = \text{ARCTAN2}\ (x\text{-}0.3333,\ y\text{-}0.3333)\cdot180/\Pi + 180 \quad (12)$$

where the ARCTAN2 function denotes a four-quadrant inverse function that allows a range of 0°–360°.

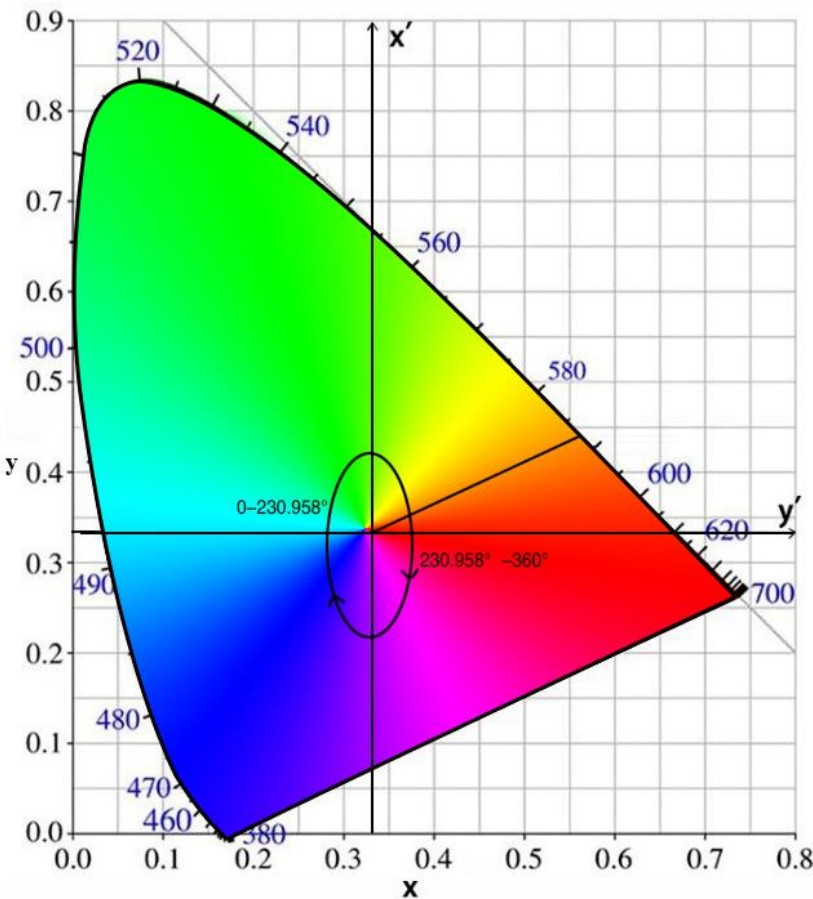

**Figure 5.** The International Commission on Communication (CIE)-xy chromaticity diagram and two kinds of water bodies. Hue angle $\alpha$ is the angle between the vector to a point and the negative x'-axis (at y = 1/3).

To prove the stability of the hue angles of the water bodies, the hue angles calculated using the quasi-synchronous Sentinel-2 image of Qiqihar City on October 24, 2018 were compared with the

hue angles calculated through in situ remote-sensing reflectance $R_{rs}(\lambda)$. First, the hue angles were extracted using remote-sensing reflectance of Sentinel-2 images [13]. Thereafter, the hue angles were extracted using in situ remote-sensing reflectance [47]. Figure 6 shows the scatterplot of the accuracy evaluation of these hue angles. The accuracy evaluation results of the hue angles were obtained as follows: $R^2 = 0.9029$, RMSE = 4.397°, and MRE = 1.744%.

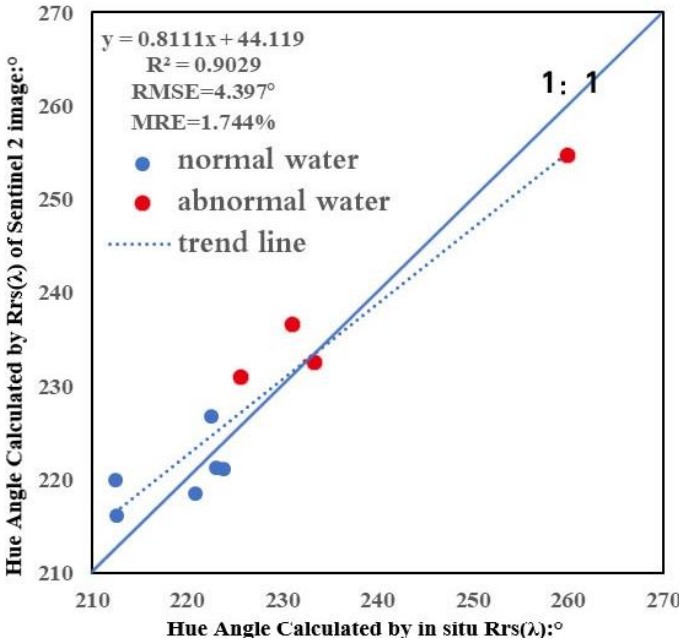

**Figure 6.** Scatterplots showing the derivation accuracies of hue angle α from remote-sensing reflectance of the Sentinel-2 image in comparison with in situ remote-sensing reflectance.

### 3.4. Recognition of Water Colour Anomaly by Using the Hue Angle of the Water Body

Conventional water-quality monitoring depends on biochemical indicators, such as the pH of a water body, involvement of heavy metals, presence of organic matter such as waste diesel, and the presence or absence of a pungent smell. As these biochemical indicators cannot be obtained from remote-sensing images, this study proposed to recognise water colour anomalies according to the hue angle of the water body. As a parameter to evaluate the water quality, the hue angle is applied to judge the water quality based on the colour of the water body. The recognition of water colour anomalies based on the hue angles of water bodies cannot only quantify water quality but also avoid the errors caused by artificial discrimination. The smaller the hue angle of the water body, the cleaner is the water body, and the larger the hue angle, the more turbid is the water body. However, no purple water body was found when the field experiments were conducted in three areas. Human vision can only identify three types of changes in colour: brightness, hue, and saturation. When the water body is grey or black, most of the light is absorbed by the water body owing to high turbidity, resulting in low brightness of the water body. Human vision is not sensitive to the brightness change of water colour, and thus, the colour of the water body observed by human eyes is grey, brown, or black, even though the actual colour of the water is between yellowish-green and red.

With the aim of defining the critical value of the hue angles of water bodies for recognising water colour anomalies, ten measured water samples in Qiqihar were taken as experimental samples; these included six general water samples and four anomalous water samples. As shown in Figure 7, $R_{rs}(\lambda)$ = 1–6 are the values for general water bodies, whereas $R_{rs}(\lambda)$ = 7–10 for water colour anomalies. The red band of water bodies with $R_{rs}(\lambda)$ = 8 and 9 is higher than that of the general water bodies, while the change in $R_{rs}(\lambda)$ = 7 and 10 is relatively smooth without obvious changes in characteristics.

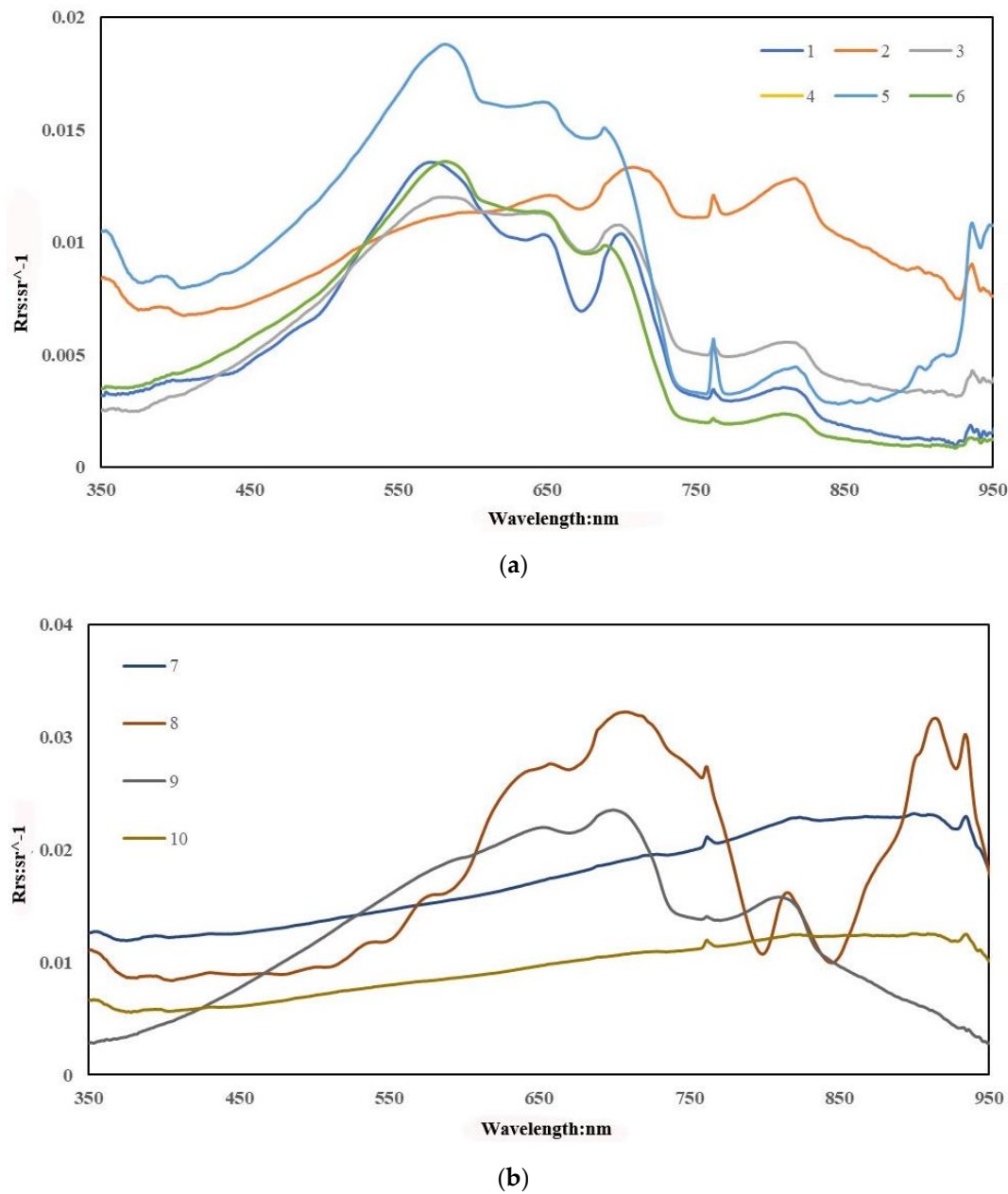

**Figure 7.** Remote-sensing reflectance of ten measured water bodies: (**a**) six general water bodies and (**b**) four water colour anomalies.

The water bodies were divided into two types according to their hue angle; i.e., when the hue angle was between 0° and 230.958°, the body was recognised as a general water body, and when the hue angle was greater than 230.958°, it was recognised as a water colour anomaly. The hue angle of the water body was calculated according to the in situ remote-sensing reflectance of ten water samples, the results of which are shown in Figure 8. In light of the above-mentioned analysis, 230.958° is taken as the critical value for recognizing a water colour anomaly.

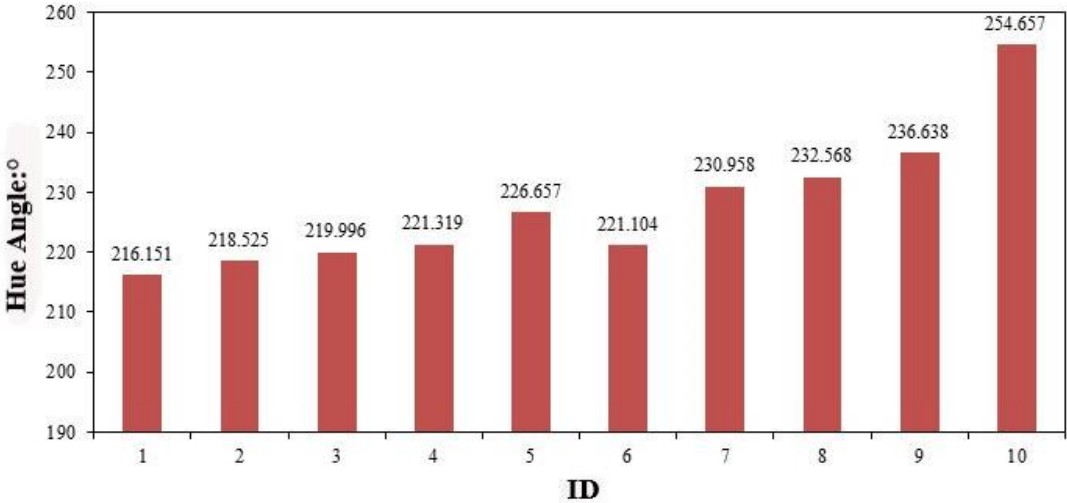

**Figure 8.** Result of the hue angles of ten water bodies in Qiqihar City.

## 4. Results

The quasi-synchronous experimental data collected on April 28, 2017 and synchronous satellite-to-ground experimental data collected on October 5, 2018 in the Xiong'an New Area were used to evaluate the accuracy of the proposed model. A total of 19 points, including three water colour anomalies and 16 general water bodies, were determined. The water colour anomalies were recognised using the critical value of 230.958°, with a recognition accuracy of 100%, thereby demonstrating that the proposed model has good recognition accuracy.

## 5. Discussion

### 5.1. Verification of the Hue Angle Threshold of a Water Body

When the colour of a water body is blue or green, it is generally recognised as a normal water body; when its colour is dark brown, greyish brown, or red, it must be a water colour anomaly; and if the colour is yellow, it is not easy to determine whether it is a water colour anomaly. Obviously, although the water bodies of the Yellow and Yangtze Rivers are yellow in colour, they are both general water bodies. Thus, it is crucial to properly select the threshold of hue angle so that these water bodies will not be misjudged as water colour anomalies. As shown in Figure 9, the water body is yellowish-green to reddish-brown when the hue angle is between 210.5766° and 238.9592° [48]. Additionally, the Sentinel-2 image data of the Wuhan section of the Yangtze River collected on October 27, 2018 and the Zhengzhou section of the Yellow River collected on August 15, 2019 were selected, and the mean value of the water pixels in homogeneous areas was calculated. This study determined that the hue angles of the Wuhan and Zhengzhou sections were 213.0879° and 220.5668°, respectively, both of which are smaller than the threshold of 230.958° set in this research. Although the eutrophic water body is green, it is not considered a water colour anomaly. The eutrophic water body in the image of the Xiong'an New Area captured on April 28, 2017 showed a hue angle of 212.53768°, which is smaller than the set threshold. Thus, the hue-angle threshold was considered appropriate, using which, the general water bodies similar to the Yellow and Yangtze Rivers, as well as eutrophic water bodies, will not be misjudged as water colour anomalies.

### 5.2. Recognition of Water Colour Anomalies in the Xiong'an New Area

Generally, the recognition method of a water colour anomaly was determined based on the measured data of Qiqihar. To further prove that the recognition method and the proposed threshold are universal, a total of 19 water samples with different colours were selected in Rongcheng County on April 28, 2017 and Anxin County of the Xiong'an New Area on October 5, 2018. A homogeneous area of

3 × 3 was selected near each water sample point to avoid the influence of shadow and optical shallow water. The hue angles of these water bodies were calculated through the RGB conversion method, the mean value of which in each selected water sample point was also calculated. The mean hue angles of the 19 water samples were obtained as 212.6984°, 198.4476°, 248.2928°, 211.9023°, 198.5374°, 284.9683°, 222.0°, 202.5556°, 199.2°, 199.0°, 165.2051°, 266.3568°, 197.0°, 171.4628°, 145.6667°, 204.1727°, 199.8333°, 165.75°, and 211.3333°. Among these 19 water body samples, the colours of three water bodies were anomalous, and the results of 16 general water samples were consistent with that of the in situ field survey, showing good recognition accuracy in the Xiong'an New Area. Hence, it is shown that the proposed method of the hue-angle threshold has certain universality. In addition, the Xiong'an New Area comprises several eutrophic water bodies; however, they were not recognised as water colour anomalies by the proposed method.

Anomalous Water

RGB

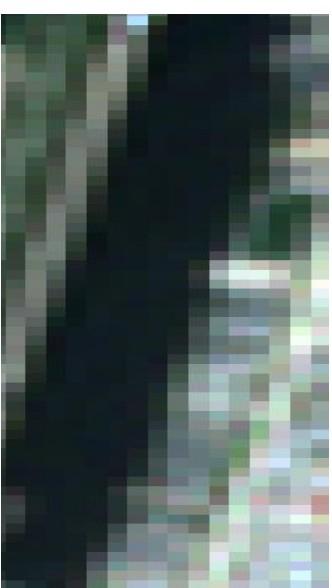

Scene
Photo

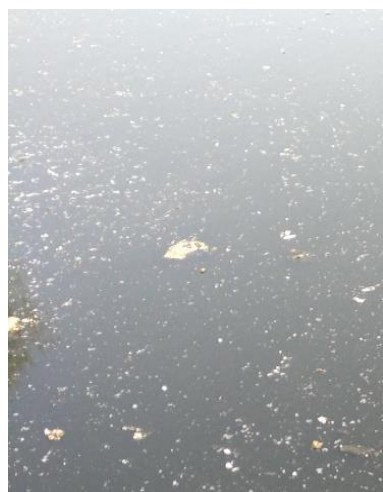

**Figure 9.** *Cont.*

$Rrs(\lambda)$

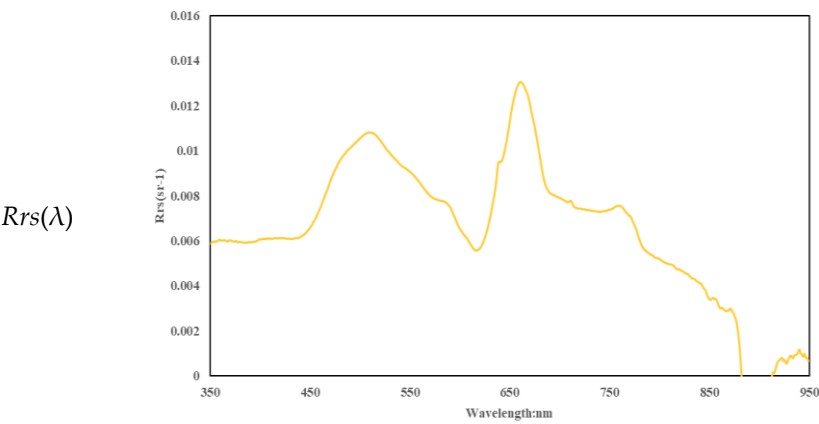

Anomalous Water

RGB

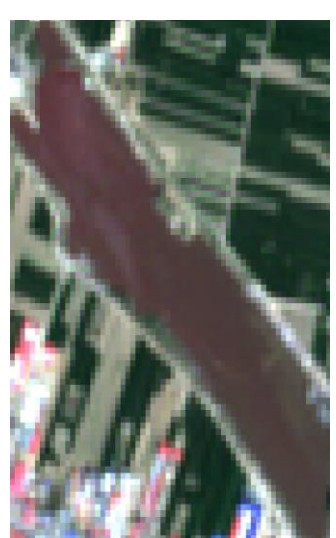

Scene
Photo

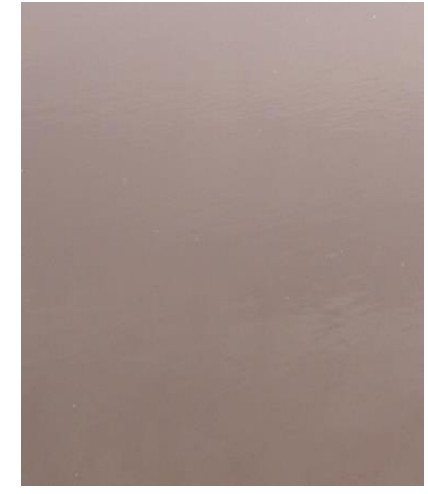

**Figure 9.** *Cont.*

$Rrs(\lambda)$

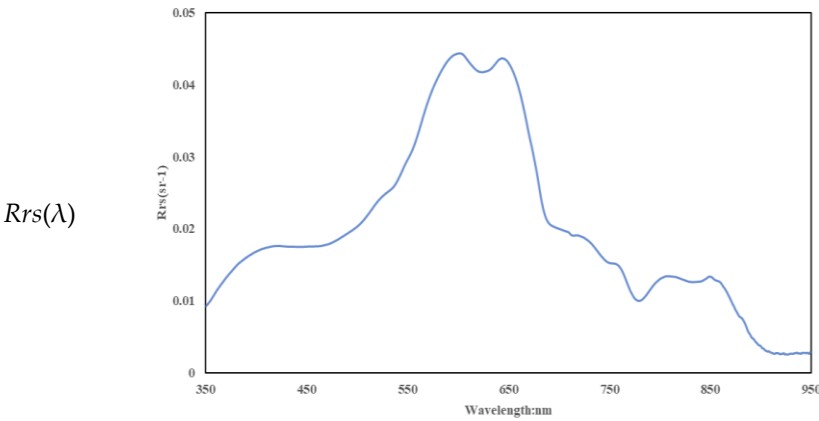

Eutrophic Water

RGB

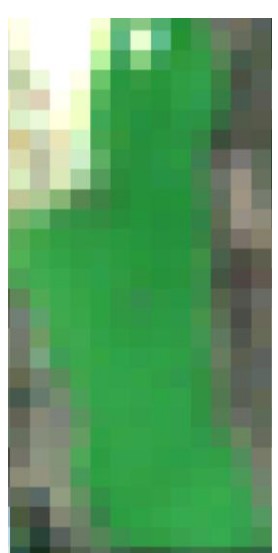

Scene
Photo

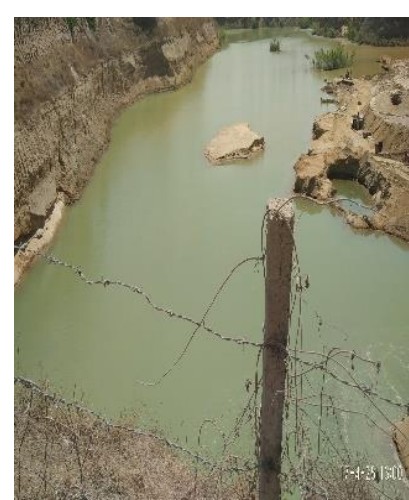

**Figure 9.** *Cont.*

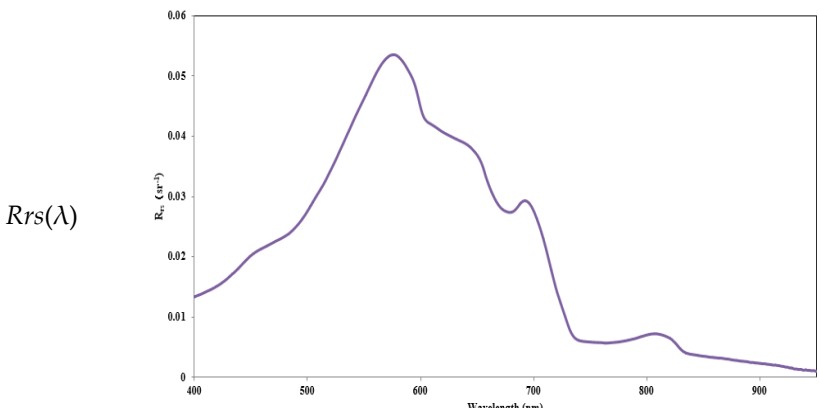

**Figure 9.** RGB synthetic image, field photo, and remote-sensing reflectance of general water bodies and water colour anomalies in the Xiong'an New Area.

### 5.3. Reasonability Evaluation of the Proposed Recognition Method for Water Colour Anomaly

Actually, most inland water bodies are ordinary water bodies with normal colour, and only a few are anomalous waters with unusual colours. In order to validate this fact, we selected a fixed time in the Xiong'an New Area (range shown in Figure 3b) on April 18, 2017 and calculated hue angles of all surface water bodies. As shown in Figure 10, in the study area of the Xiong'an New Area, 91% of the monitored water bodies are general water bodies, including eutrophication water bodies, and the hue angle of each water body is less than 230.958°; when the hue angle of the water body is more than 230.958°, it falls under water colour anomalous waters, and the number of such water bodies accounts for only 9% of all inland water bodies in the area. These results show that most of the inland water bodies are general water bodies; that is, anomalous waters account for only a small part of the inland water bodies. The results of this method are basically consistent with this fact. Therefore, the results of this method are reasonable and consistent with the actual spatial distribution, which proves the feasibility of this method to a certain extent.

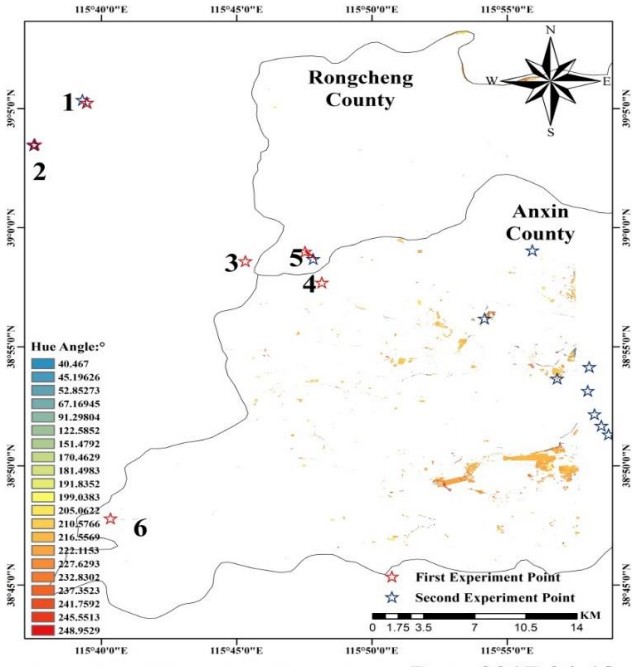

**Figure 10.** The results of the hue angles of the water bodies in the Xiong'an New Area.

The first assumption here is that in the inland water bodies, such as lakes, rivers, ponds, and other general water bodies, the colours of the water bodies cannot stay the same in different periods. They will change with time; these changes are subtle and will increase dramatically only under the interference of external factors. Similarly, the second assumption here is that, for waters with unusual colours, if there is no treatment of some artificial influence, the colours of the water bodies will not change dramatically but gradually. To prove the above two assumptions, we selected water body No. 5 (a general water body) and water body No. 8 (an anomalous water) in Qiqihar. Note that the water bodies we selected are in Qiqihar (range shown in Figure 3a), not in the Xiong'an New Area. We selected 4 × 4 pixels at the centre of the water area and calculated the water hue angle of the area in April 2016–August 2019. The 4 × 4 area at the centre of the water body was chosen because the water near the shore is often shallow optical water, and the colour of the water body will be significantly affected by the bottom material. The area at the centre of the lake or pond will be less affected; in addition, the average value of 4 × 4 can reduce the influence of noise compared with selecting a single pixel. It can be seen from Figure 11 that the hue angles of water body No. 5 changed slightly with time in each month. The hue angle of water body No. 8 also changed slightly. In fact, the hue angle of water body No. 8 changed slightly without any governing measures or the interference of external factors (such as precipitation, industrial pollution, marine/fluvial litter, etc). The first assumption and the second assumption correspond to water body No. 5 and water body No. 8, respectively. The results obtained using this method are basically consistent with the two assumptions, which further proves that our method has good stability.

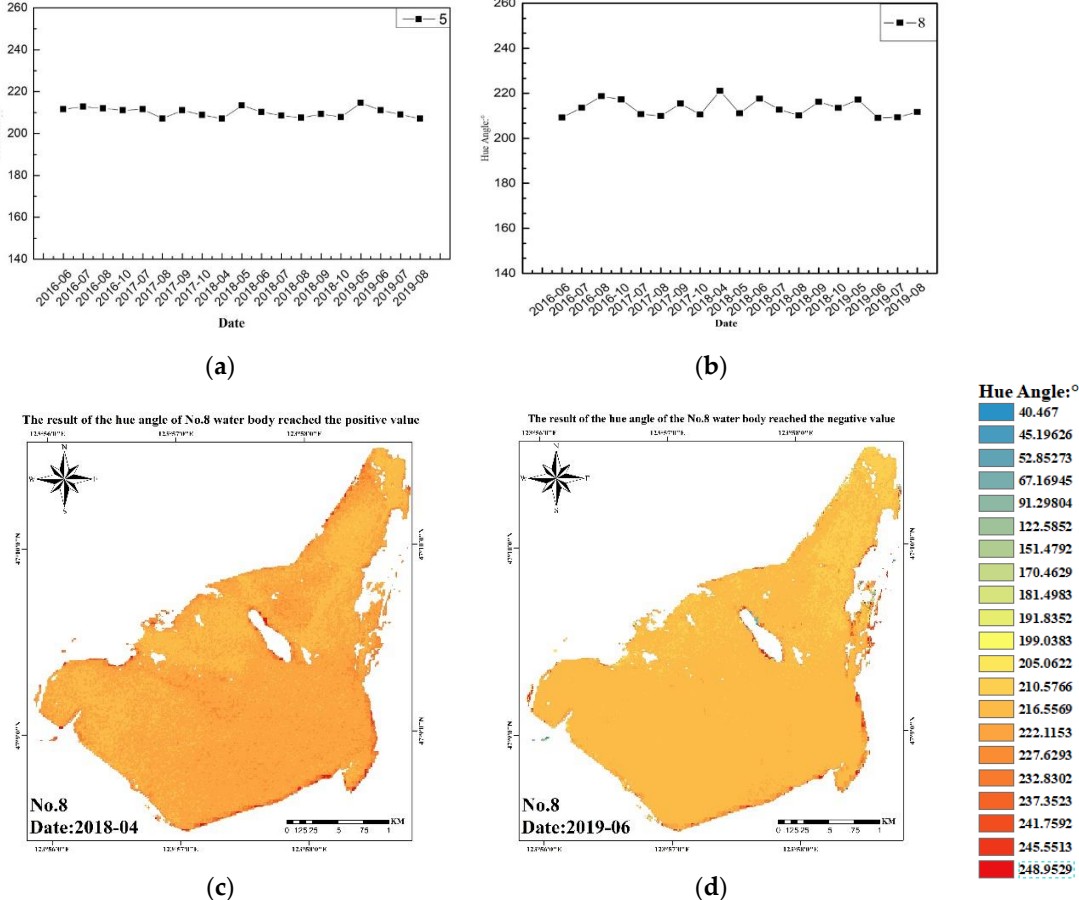

**Figure 11.** Statistics of water body hue angle changes in (**a**) water body No. 5 and (**b**) water body No. 8 in Qiqihar City from May 2016 to October 2019. (**c**) The highest hue angle of water body No. 8 was reached in April 2018. (**d**) The lowest hue angle of the same water body was reached in June 2019.

In addition, the method proposed is suitable for optical deep water but not for optical shallow water. The water bodies in the two research areas were optical deep waters. Indeed, in several areas of China, the inland water bodies are optical deep waters. The transparency of these water bodies is far less than the depth of the water bodies. The method proposed in this paper is therefore applicable in several inland water areas of China.

## 6. Conclusions

The following are the main conclusions drawn from this study.

(1) The extraction results based on MuWI were evaluated through MRE and RMSE of 0.04% and 7.48°, respectively.

(2) The hue angles of water bodies calculated by the quasi-synchronous Sentinel-2 images were compared with the hue angles calculated through in situ remote-sensing reflectance, $R_{rs}$ ($\lambda$). The accuracy evaluation results of the hue angles were obtained as follows: $R^2 = 0.9029$, RMSE = 4.397°, and MRE = 1.744%.

(3) Based on the in situ $R_{rs}$ ($\lambda$) of the water body in Qiqihar, the equivalent $R_{rs}$ ($\lambda$) corresponding to the Sentinel-2 image band was simulated, and the hue angles of the water bodies were calculated. Owing to the differences between the hue angles of the general water bodies and water colour anomalies, the threshold of the hue angle for recognising water colour anomalies was set at 230.958°. As such, an algorithm based on Sentinel 2 was proposed to recognise water colour anomalies.

(4) The hue angle in the CIE space is a crucial colour parameter. This study is the first to apply this parameter to recognise water colour anomalies, with a good recognition accuracy. The hue angle of a water body can be well applied to Sentinel-2 images and, therefore, has good recognition accuracy.

(5) By calculating the results of multi-scene images from 2016 to 2019 and the monthly mean value of the hue angle of each water body, the method was proved to show good stability and universality.

(6) This paper put forward a method of recognizing water colour anomalies according to the hue angles of water bodies; this can be applied to water bodies with red, grey, or black colours regarded as water colour anomalies but not to eutrophic water bodies with a green colour. Moreover, the method proposed is only suitable for optical deep waters and not for optical shallow waters.

**Author Contributions:** Conceptualization, Y.Z.; methodology, Y.Z. and S.W.; validation, F.Z., Q.W., Y.Y., and F.Y.; formal analysis, Q.S., S.W., and J.L.; writing—original draft preparation, Y.Z.; writing—review and editing, Q.S. and J.L.; and funding acquisition, Q.S. All authors have read and agreed to the published version of the manuscript.

**Funding:** This work was supported by the National Key Research and Development Program (2017YFB0503902), National Natural Science Foundation of China (Grant No. 41571361), and Strategy Priority Research Program Project of the Chinese Academy of Sciences (Grant No. XDA23040102).

**Acknowledgments:** The authors would like to thank ESA for providing the Sentinel-2 data, thank Qian Shen from the Key Laboratory of Digital Earth Science, Aerospace Information Research Institute, Chinese Academy of Sciences for the help with writing, and thank Shenglei Wang from the PKU Institute of Remote Sensing and Geographical Information System, Peking University for the help during the Sentinel-2 data batch processing. We would like to thank Editage (www.editage.cn) for English language editing.

**Conflicts of Interest:** The authors declare no conflicts of interest.

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
