# Peer review of "Recognition of Water Colour Anomaly by Using Hue Angle and Sentinel 2 Image"

_remotesensing, doi:10.3390/rs12040716_

Round 1

Reviewer 1 Report

Dear Authors,

I enjoyed reading through your paper, and have mostly technical and grammatical edits.

Best wishes.

Major fixes

Figure 2 is cut across a page break, I will leave that up to the editors if that is acceptable Map points on Figure 3 are hard to see against the basemap. Blue stars on Figure 3(b) very hard to see when paper is printed out. I suggest changing the colors to something brighter Section 2.2 Data Acquisition is in the paper twice, as well as Figure 3 (labeled Figure 5 in the reposting).  There is a captain for a figure 4, but there is no figure 4. The captain for figure 4 is the same as figure 3. Removing Figures 4 and 5 from the paper will cause subsequent figure references in the text to be renumbered Figure 11 is cut off by a page break. Also, the captain "Anomalous Water" is cutting off and wrapping the word "Anomalous" Figure 12 is cut off by a page break. Make the rows smaller so that it will fit on an entire page like figure 13. The last row of figure 12 is not lined up with the rest of the figure.

Minor Fixes:

Intro:

I would replace "(3) aquaculture" with "agriculture." You are using this category to describe runoff from agricultural practices. aquaculture is the growing of fish in captivity for human consumption, and does not fit the context used in the text. in the paragraph following figure 1, midway through remove the "the" after "Conversely" so that the sentence reads "Conversely, remote-sensing technology has..." in the top paragraph of the 3rd page, there should be a space in "WilliUle"

Study Area and Data:

In the 1st paragraph, the second sentence that contains reference [36], you need to adjust the tense of the words used. It needs to either say "....formed a huge black and odorous water body." OR "formed huge black and odorous water bodies." the "a" in that sentence does not agree with the plural "bodies"

In Situ Remote-Sensing Reflectance

The first paragraph, mid-way through has "reference board" listed twice in the sampling order. I was not certain if that was intentional or a typo.

Body Water Extraction

last sentence before section 3.2, "indexes" should be "indices." The paper has "indexes" many other times as well, so be on the lookout and update.

Recognition of water color anomaly by using the hue angle of water body

i would replace "acidity and basicity" with "pH"

Accuracy Evaluation of water extraction

first sentence has "indexes" again. replace with "indices" second paragraph, first sentence has "indexes" last sentence in the 3rd paragraph, change "outline" to "outlined". "....the water body was artificially outlined...."

Recognition of water color anomaly in Xiong'an New Area

Verb tense is off on the 6th line of the first paragraph. "The hue angle of these water bodies were calculated...." Either change "angle" to "angles" or "were" to "was." "The hue angles of these water bodies were calculated...." "The hue angle of these water bodies was calculated..." Second to last sentence, i suggest changing "Hence, it is proved" to "Hence, it is shown"

Stability evaluation of the proposed recognition method for water color anomaly

Suggest changing second sentence from "With the aim of proving that...." to "With the aim of demonstrating that..." Sentence at the end of the 4th line begins with "In this study...." A number of sentences begin with this exact phrase. I suggest going back through the paper and rewording some of them so it doesn't come off as repetitive. The second to last sentence of this paragraph "were inclined be stable." needs to be reworded. perhaps "were inclined to be stable."

Author Response

Author’s Response to Decision Letter (Remote Sensing-701672) Recognition of Water Color Anomaly by using Hue Angle and Sentinel 2 Image ________________________________________ All changes in the manuscript are indicated in yellow. All responses are in red. Response of Comments of Reviewer 1: Several special comments: Point 1: Figure 2 is cut across a page break, I will leave that up to the editors if that is acceptable. Map points on Figure 3 are hard to see against the basemap. Blue stars on Figure 3(b) very hard to see when paper is printed out. I suggest changing the colors to something brighter. Section 2.2 Data Acquisition is in the paper twice, as well as Figure 3 (labeled Figure 5 in the reposting). There is a captain for a figure 4, but there is no figure 4. The captain for figure 4 is the same as figure 3. Removing Figures 4 and 5 from the paper will cause subsequent figure references in the text to be renumbered. Figure 11 is cut off by a page break. Also, the captain "Anomalous Water" is cutting off and wrapping the word "Anomalous". Figure 12 is cut off by a page break. Make the rows smaller so that it will fit on an entire page like figure 13. The last row of figure 12 is not lined up with the rest of the figure. Response 1: We have corrected this mistake in the revised manuscript. Both a and b figures in Figure 2 are on the same page. The blue stars in Figure 3(b) has been changed to green. Figure 3 (b) Xiong’an New Area. All figures in this paper are renumbered in the revised manuscript. Figure 5 in the previous article is figure 6 in the current article, and the color of the histogram has been modified to change from blue to red. Figure 8 Result of hue angle of 10 water bodies in Qiqihar City. Each figure is adjusted on an entire page. The captain "Anomalous Water" of Figure 9 has been corrected. Figure 11 and 12 in the last version are replaced with Figure 10 and 11 in the current version. Some typographical errors in the previous version may be due to different display effects of word on different machines. To avoid such mistakes, we will upload the pdf version this time. Point 2: Intro:I would replace "(3) aquaculture" with "agriculture." You are using this category to describe runoff from agricultural practices. aquaculture is the growing of fish in captivity for human consumption, and does not fit the context used in the text. In the paragraph following figure 1, midway through remove the "the" after "Conversely" so that the sentence reads "Conversely, remote-sensing technology has...". In the top paragraph of the 3rd page, there should be a space in "WilliUle" Response 2: We have corrected these mistakes in the revised manuscript. Page 2, the 59th line and the 65th line. Point 3: Study Area and Data: In the 1st paragraph, the second sentence that contains reference [36], you need to adjust the tense of the words used. It needs to either say "....formed a huge black and odorous water body." OR "formed huge black and odorous water bodies." the "a" in that sentence does not agree with the plural "bodies" Response 3: We have corrected these mistakes in the revised manuscript. Modified to "formed huge black and odorous water bodies. " Page 4, the 150th line. Point4: In Situ Remoe-Sensing Reflectance:The first paragraph, mid-way through has "reference board" listed twice in the sampling order. I was not certain if that was intentional or a typo. Response 4: The operation is not typo. The reference board is measured successively to check whether the solar radiation is stable or not. Point 5: Body Water Extraction last sentence before section 3.2, "indexes" should be "indices." The paper has "indexes" many other times as well, so be on the lookout and update. Response 5: We have replaced all "indexes" in the revised manuscript with "indices". Point 6: Recognition of water color anomaly by using the hue angle of water body i would replace "acidity and basicity" with "pH" Response 6: We have replaced "acidity and basicity" with "pH" in the revised manuscript. Page 11, the 309th line. Point 7: Accuracy Evaluation of water extraction: first sentence has "indexes" again. replace with "indices" second paragraph, first sentence has "indexes" last sentence in the 3rd paragraph, change "outline" to "outlined". "....the water body was artificially outlined...." Response 7: We have corrected these mistakes in the revised manuscript. Page8, last line. Point 8: Recognition of water color anomaly in Xiong'an New Area:Verb tense is off on the 6th line of the first paragraph. "The hue angle of these water bodies were calculated...." Either change "angle" to "angles" or "were" to "was." "The hue angles of these water bodies were calculated...." "The hue angle of these water bodies was calculated..." Second to last sentence, i suggest changing "Hence, it is proved" to "Hence, it is shown" Response 8: We have corrected these mistakes in the revised manuscript. Page 14, the 381th line and the 388th line. Point 9: Stability evaluation of the proposed recognition method for water color anomaly:Suggest changing second sentence from "With the aim of proving that...." to "With the aim of demonstrating that..." Sentence at the end of the 4th line begins with "In this study...." A number of sentences begin with this exact phrase. I suggest going back through the paper and rewording some of them so it doesn't come off as repetitive. The second to last sentence of this paragraph "were inclined be stable." needs to be reworded. perhaps "were inclined to be stable." Response 9: We have replaced "With the aim of proving that... " with "With the aim of demonstrating that... ". Page 15, the 397th line. We go back through the paper and rewording all "in this study" so it doesn’t come off as repetitive. Page 1, the 35th line with "In this paper"; Page 7, the 202th line with "In this research"; Page 7, the 217th line with "Generally"; Page 14, the 364th line with "And"; Page 14, the 369th line with "In this research"; Page 14, the 376th line with "Generally". Other two "in this study" were deleted on the 226th line and 399th line. The second to last sentence of this paragraph, "were inclined to be stable." replace "were inclined be stable.". Page 16, the 434th line.

Reviewer 2 Report

I reviewer the manuscript entitled 'Recognition of Water Color Anomaly by using Hue Angle and Sentinel 2 Image' submitted to the MDPI Remote Sensing journal.

As the PDF file is not numbered (line numbers are missing) I will give indications by referring to n of pages of based on the position of the text respect to figures or equations or by referring to whole sections or sub-sections.

The following major revisions need to be addressed before the manuscript can be published.

Section 2.2 is reported two times during the manuscript.

In section 3.3 from the text from 'With the aim of defining the critical value...' to Fig. 8 is to be moved in results section. This preparatory work need to be described in section 3.3 briefly only without reporting also the associated results.

Similarly, in sections 4.1, 4.2 and 4.3 equations and associated description need to be moved in methods. Section 4 could be reports only results.

Fig.10 need to report in x- and y- axes a label which allows to identify what are the Hue angles from Sentinel and those computed using in situ Rrs.

Caption of Fig 10 reports the term 'hyperspectral integration', what it is mean? this is never mentioned before. Are the authors referring to some manipulation of in situ spectral reflectance sempling?

In section 5.3 the authors refer to different atmospheric composition if a a time series of images is used. This could be already taken into account by applying the atmospheric correction. Thus, i don't see the link with the following sentence.

In the further analysis it is not clear why the hue angle was monthly aggregated...it is only to discuss a map per month?

Fig 12 and 13 and the section 5.3 could be the core of the manuscript. The text in section 5.3 sounds like an abstracted result...the authors need to expand a lot this section.

The two figures (fig 12 and 13) are really hard to interpret as the water bodies are several and very small in the figures. So, it is hard to the reader compare month by month the averaged hue angle and extract from these figures the main outcomes of this research.  Maybe the authors can focused on the 2D representation of a less number of water bodies by selecting those in which results are the more relevant. For all water bodies the authors can show a graph of the hue angle (average and st.dev of each water body) with time and discuss this graph.

Finally, this research is based on the analysis of 'color index' retrieved by Remote sensing reflectance of water bodies to be related to water quality...

in this framework, as reviewer i need to point out that your study is valid only if variations of the color index are indipendent from the sea bed (or river bed) spectral reflectances.

Indeed, Rrs for shallow water (less than 30-40 m) is for sure dependent also from the spectral reflectance of the bed; this becoming of less importance as the water is less transparent (polluted waters, high-eutrophic water environments, waters characterized by high turbidity...). This point could be addressed by analyzing in depth the in situ observation of Rrs.

Author Response

All changes in the manuscript are indicated in yellow. All responses are in red.

Response to Comments of Reviewer 2:

Point 1:As the PDF file is not numbered (line numbers are missing) I will give indications by referring to n of pages of based on the position of the text respect to figures or equations or by referring to whole sections or sub-sections.

Response 1: We are sorry for the inconvenience we had made. We have added line numbers to the revised manuscript.

Point 2: The following major revisions need to be addressed before the manuscript can be published.

Section 2.2 is reported two times during the manuscript.

Response  2: Some typographical errors in the previous version may be due to different display effects of word on different machines. To avoid such mistakes, we will upload the pdf version this time.

Point 3: In section 3.3 from the text from 'With the aim of defining the critical value...' to Fig. 8 is to be moved in results section. This preparatory work need to be described in section 3.3 briefly only without reporting also the associated results.

Response 3: Thank you for your suggestion. We have made modifications as your suggestion. We arrange "accuracy evaluation indices" as a new section 3.1.  Also, the part of the accuracy evaluation of water body extraction was added into the corresponding Section 3.2 briefly. The part of the accuracy evaluation of hue angle method was added into the corresponding Section 3.3 briefly. 

Point 4: Similarly, in sections 4.1, 4.2 and 4.3 equations and associated description need to be moved in methods. Section 4 could be reports only results.

Response 4: Similarly, in sections 4.1, 4.2 and 4.3 equations and associated description have been to be moved in methods. Now Section 4 in the revised manuscript report only the results of the hue angle algorithm identifying waters with color anomalies.

Point 5: Fig.10 need to report in x- and y- axes a label which allows to identify what are the Hue angles from Sentinel and those computed using in situ Rrs.

Response 5: Figure 10 in the previous manuscript is figure 6 in the revised manuscript. We explain the coordinate axes in the paper respectively.

Round 2

Reviewer 2 Report

Thanks the authors for have provided a new manuscript release which fit all my comments/suggestions

However the manuscript in considerably my opinion is now improved,a silgtly effort it is need to improve the result part. 

In particular, in the v1 release fig 12 and 13 were difficult to read, authors replaced these two figures with fig 10 and 11. 

Fig 10 is i correctly understood refer to the whole area shown in fig 3 b. however, it could be useful to add the numbering of the monitored sub-areas also in fig 10 (as done in fig 3).

In addition, Fig 10 now shows only a fixed time. It could be useful for sub-area 8 show some images in time and also to discuss the observed changes in time...are these changes due to some well know problem in the area?  turbidity after rainfall? industries pollution? marine/fluvial litter?

Images of the water body n8 could be shown at the time in which in fig 11 positive/negative peaks occur.

Author Response

Author’s Response to Decision Letter (Remote Sensing-701672)

Recognition of Water Color Anomaly by using Hue Angle and Sentinel 2 Image

All changes in the manuscript are indicated by yellow highlighting. All responses below are in red.

Response to Comments of Reviewer 2:

Several special comments:

Point 1: Fig 10 is i correctly understood refer to the whole area shown in fig 3 b. however, it could be useful to add the numbering of the monitored sub-areas also in fig 10 (as done in fig 3).

Response 1: Thank you for this comment, with which we agree. We have added the numbering of the monitoring areas also in Figure 10 (as done in Figure 3).

Figure 10. The result of the hue angle of water bodies in Xiong’an New Area.

Point 2: In addition, Fig 10 now shows only a fixed time. It could be useful for sub-area 8 show some images in time and also to discuss the observed changes in time...are these changes due to some well know problem in the area?  turbidity after rainfall? industries pollution? marine/fluvial litter?

Response 2: Thank you very much for your comments and suggestions. Actually, most inland water bodies are ordinary water bodies with normal colour, and only a few are anomalous waters with unusual colours. Figure 10 shows a fixed time that is calculated to show that the anomalous waters account for only a small part of the inland water bodies. In addition, water body No. 8 in the second paragraph of Section 5.3 is in Qiqihar City, not in Xiong’an New Area covering Figure 10, which is not the same area as Figure 10. The reason why we calculated the water hue angle of water body No. 8 in April 2016 to August 2019 is to prove the second assumption in the second paragraph of Section 5.3, that is, the hue angle changes slightly even when there are no governing measures or interference of external factors (such as precipitation, industrial pollution, marine/fluvial litter, etc.).

Point 3: Images of the water body n8 could be shown at the time in which in fig 11 positive/negative peaks occur.

Response 3: We fully agree with you. In the revised manuscript, we have shown in Figure 11 the times at which the highest and lowest peaks occur for water body No. 8. In April 2018, the hue angle of water body No. 8 reached its highest positive peaks. In June 2019, the hue angle of water body No. 8 reached its lowest values. It can be seen that there is little difference in colors between the two pictures.

  • (b)

(c)                               (d)

Figure 11. Statistics of water body hue angle changes in (a) water body No. 5 and (b) water body No. 8 in Qiqihar City from May 2016 to October 2019. (c) The highest hue angle of water body No. 8 reached in April 2018. (d) The lowest hue angle of the same water body reached in June 2019.
